# Hyperoxic BOLD-MRI-Based Characterization of Breast Cancer Molecular Subtypes Is Independent of the Supplied Amount of Oxygen: A Preclinical Study

**DOI:** 10.3390/diagnostics13182946

**Published:** 2023-09-14

**Authors:** Silvester J. Bartsch, Viktoria Ehret, Joachim Friske, Vanessa Fröhlich, Daniela Laimer-Gruber, Thomas H. Helbich, Katja Pinker

**Affiliations:** 1Department of Biomedical Imaging and Image-Guided Therapy, Division of Structural and Molecular Preclinical Imaging, Medical University of Vienna, 1090 Vienna, Austria; silvester.bartsch@meduniwien.ac.at (S.J.B.); joachim.friske@meduniwien.ac.at (J.F.); daniela.laimer-gruber@meduniwien.ac.at (D.L.-G.); thomas.helbich@meduniwien.ac.at (T.H.H.); 2Department of Internal Medicine III, Division of Endocrinology and Metabolism, Medical University of Vienna, 1090 Vienna, Austria; viktoria.ehret@meduniwien.ac.at; 3Fachhochschule Wiener Neustadt GmbH, University of Applied Sciences, 2700 Wiener Neustadt, Austria; vanessa.froehlich@fhwn.ac.at; 4Breast Imaging Service, Department of Radiology, Memorial Sloan Kettering Cancer Center, New York, NY 10065, USA

**Keywords:** breast cancer characterization, hypoxia, hyperoxic BOLD-MRI, intrinsic contrast MRI

## Abstract

Hyperoxic BOLD-MRI targeting tumor hypoxia may provide imaging biomarkers that represent breast cancer molecular subtypes without the use of injected contrast agents. However, the diagnostic performance of hyperoxic BOLD-MRI using different levels of oxygen remains unclear. We hypothesized that molecular subtype characterization with hyperoxic BOLD-MRI is feasible independently of the amount of oxygen. Twenty-three nude mice that were inoculated into the flank with luminal A (*n* = 9), Her2^+^ (*n* = 5), and triple-negative (*n* = 9) human breast cancer cells were imaged using a 9.4 T Bruker BioSpin system. During BOLD-MRI, anesthesia was supplemented with four different levels of oxygen (normoxic: 21%; hyperoxic: 41%, 71%, 100%). The change in the spin–spin relaxation rate in relation to the normoxic state, $\Delta R_{2}{}^{*}$, dependent on the amount of erythrocyte-bound oxygen, was calculated using in-house MATLAB code. $\Delta R_{2}{}^{*}$ was significantly different between luminal A and Her2^+^ as well as between luminal A and triple-negative breast cancer, reflective of the less aggressive luminal A breast cancer’s ability to better deliver oxygen-rich hemoglobin to its tissue. Differences in $\Delta R_{2}{}^{*}$ between subtypes were independent of the amount of oxygen, with robust distinction already achieved with 41% oxygen. In conclusion, hyperoxic BOLD-MRI may be used as a biomarker for luminal A breast cancer identification without the use of exogenous contrast agents.

## 1. Introduction

Hypoxia, i.e., the lack of oxygen within tissue, has been identified as one of the most pervasive tumor microenvironment factors in cancer progression [1,2,3,4]. Hypoxia induces aggressive and treatment-resistant tumor phenotypes, leading to rapid tumor progression and poor prognosis [5]. In breast cancer, hypoxia has been shown to induce hallmarks of cancer including angiogenesis (“angiogenic switch”) and the deregulation of cell metabolism (“metabolic switch”) [6,7,8]. Specifically, the response to hypoxia on a cellular level is mediated by the expression of so-called hypoxia-inducible factors (e.g., HIF1-α) [1,2,4]. As a result, under persisting hypoxic conditions, HIF1-α accumulates within cells and eventually translocates into the nucleus to induce the expression of vascular endothelial growth factors (VEGFs) [9,10]. The presence of VEGFs promotes the activation of blood-vessel inducers, switching on angiogenesis. The distribution of these key adaptive hallmarks of cancer progression is not uniform within a given primary tumor, resulting in intra-tumoral heterogeneity. Of note, breast cancer is a disease with remarkable intra-tumor heterogeneity where multiple cell populations co-exist within the primary tumor [11,12,13]. Currently, however, treatment decisions for breast cancer are based on histological, immunohistochemical, and molecular profiling by invasive tissue sampling of small tumor subregions, which are not necessarily representative of the entire tumor. The treatment plans that are based on invasive tissue sampling from breast biopsies that are inherently prone to site selection bias may not be representative of the tumor in its entirety, which can be an important contributing factor for disease recurrence in up to 30% of breast cancer patients [14]. To improve treatment outcome, deciphering hypoxia-induced breast cancer heterogeneity has to be accomplished prior to patient treatment. Considering this, novel imaging biomarkers targeting the entire tumor, including imaging biomarkers targeting hypoxia, are urgently needed to facilitate appropriate diagnosis and guide precision treatment. Apart from facilitating a non-invasive assessment of tumor physiology, imaging biomarkers will enable a holistic view on tumor heterogeneity induced by the hypoxic tumor microenvironment.

Magnetic resonance imaging (MRI) offers insights into the oxygen levels in the blood stream via the blood-oxygen-level-dependent (BOLD) effect. The BOLD effect is based on the differential magnetic properties of deoxy- and oxyhemoglobin, therefore exploiting blood as a ubiquitous intrinsic contrast agent [15,16,17]. Oxygenized hemoglobin, with its lack of unpaired electrons, is weakly diamagnetic. Conversely, in the case of deoxygenized hemoglobin, unbound binding sites of iron atoms bound to hemoglobin increase local susceptibility gradients, which induces magnetic field distortions and results in alterations in the effective spin–spin relaxation rate that can be exploited as contrast in MR imaging. The resulting so-called BOLD effect has been a key mechanism in functional brain MRI [18,19] and has proven to be a promising imaging biomarker of hypoxia [15,20]. In cancer imaging, the BOLD effect is mostly harnessed by measuring the effective spin–spin relaxation rate $R_{2}{}^{*}$ under normoxic breathing conditions (i.e., 21% O_2_), which is commonly understood as an indicator for hemoglobin-bound oxygen within the tumor. Hypoxic tumor areas are detected by an increase in $R_{2}{}^{*}$, which results from the paramagnetic susceptibility of increased deoxyhemoglobin compared to healthy tissue [3,21].

Recently, hyperoxic BOLD-MRI was introduced, where baseline measurement cycles are performed while the patient breathes regular air, followed by measurement cycles under hyperoxic conditions using either 100% O_2_ or carbogen (a mixture of 95% O_2_ and 5% CO_2_) [15,22]. The resulting decrease in $R_{2}{}^{*}$, called $\Delta R_{2}{}^{*}$, allows for precise insight into the tumor’s oxygen supply and resulting hypoxia, possibly enabling improved tumor characterization and assessment of therapy response. The rationale behind hyperoxic gas challenges is that by turning the fraction of oxygen to 100% O_2_ (pO_2_ = 760 mmHg), the amount of oxyhemoglobin reaches its maximum, resulting in maximal BOLD contrast [23,24]. However, the oxygen equilibrium curve suggests that the fraction of oxyhemoglobin reaches a plateau and only increases minimally beyond a partial oxygen pressure of pO_2_ = 70 mmHg. Indeed, at alveolar pO_2_ = 100 mmHg, which corresponds to approximately 21% O_2_ in the breathing gas (i.e., normoxic conditions), over 95% of hemoglobin is oxygenized. Furthermore, if oxyhemoglobin is nearly maximal at 21% O_2_, switching the amount of oxygen to 100% O_2_ for hyperoxic BOLD-MRI may not be necessary. In other words, a sufficient decrease in $R_{2}{}^{*}$ for the assessment of tumor oxygen provision may already be achieved at lower oxygen levels. Therefore, further refinement of hyperoxic gas challenges is necessary to pave the way for BOLD-MRI as a contrast-agent-free alternative in standard clinical practice.

We hypothesized that the molecular subtype characterization of breast cancer can be achieved with hyperoxic BOLD-MRI independently of the amount of oxygen. To this end, in this preclinical study, we investigated hyperoxic BOLD-MRI using different hyperoxic gas challenges of varying oxygen concentrations of 41%, 71%, and 100% O_2_ in characterizing xenograft tumors of different breast cancer molecular subtypes.

## 2. Materials and Methods

### 2.1. Cell Culture and Mouse Model

This preclinical study was approved by the Austrian Federal Ministry of Education, Science and Research (project number BMFWF-66.009/0284-WF/V/3b/2017) and all study procedures were conducted in accordance with the European Community’s Council Directive of 22 September 2010 (2010/63/EU).

Luminal A, Her2^+^, and triple-negative breast cancer molecular subtypes are considered less aggressive, moderately aggressive, and highly aggressive breast cancer, respectively, based on their treatment prognosis in the clinical context. Therefore, tumor cells of luminal A (MCF-7), Her2^+^ (SKBR-3), and triple-negative (MDA-MB-231) breast cancer molecular subtypes were cultivated in RPMI (MCF-7) and DMEM (SKBR-3 and MDA-MB-231) cell culture medium, supplemented with 10% fetal bovine serum and 2% penicillin/streptomycin antibiotics. All tumor cells were kept in standardized conditions at 37 °C in a humidified incubator in an atmosphere containing 5% CO_2_ during cultivation.

Subsequently, four- to six-week-old female athymic Balb/c nude mice (*n* = 28) from Charles River Laboratories (Wilmington, MA, USA) were inoculated subcutaneously into the right flank with 2 × 10^7^ luminal A (*n* = 10), Her2^+^ (*n* = 8), or triple-negative (*n* = 10) tumor cells suspended in 100 µL of serum-free growth medium using a 27 G injection needle. The tumor growth of luminal A-derived tumor xenografts was supported by subcutaneous implantation of half an estrogen pellet (0.72 mg/day, 60-day release, Innovative Research of America, Sarasota, FL, USA) in the neck area. The tumors were allowed to grow for around 1–2 weeks to reach sizes suitable for MR examination with a maximal diameter of 10 mm. Of note, the mice were anesthetized using 3.0% isoflurane (Zoetis Österreich GmbH, Vienna, Austria) mixed with breathing air at a flow rate of 2.0 L/min for induction for all experimental and imaging (detailed below) procedures, and their breathing rate maintained at approximately 60 breaths/min using 1.5–2.0% of isoflurane in the anesthetic gas.

### 2.2. MRI

MRIs were performed in all mice using a 9.4 T Bruker Biospec 94/30USR system (Bruker, Ettlingen, Germany) equipped with a BGA 12S HP gradient system combined with a 40 mm ^1^H transmit-receive volume coil (Bruker T13162V3). The system was operated using the Paravision 7 software suite (Bruker, Ettlingen, Germany). During imaging, the body temperature of the mice was kept at 37 °C using a water heating pad. The vital functions of all mice were monitored using an SA II monitoring system (SA Instruments, Stony Brook, NY, USA) with respiratory gating to limit movement artifacts. At first, a series of *T*_1_-weighted images were acquired as anatomical reference using a 2D RARE sequence with variable repetition times for *T*_1_ mapping (VTR: 1472 ms, 2.000 ms, 3.000 ms, 4.000 ms, 5.000 ms, 7.000 ms, 8.000 ms, 9.000 ms; TE: 21.0 ms, matrix size: 128 × 128; number of slices: 10; slice thickness: 1 mm; spatial resolution: 0.278 × 0.222 mm; acquisition time: 16 min). Following scout image acquisitions, $T_{2}{}^{*}$ mapping was repeated three times to acquire baseline (i.e., normoxic 21% O_2_ in the anesthetic gas) BOLD measurements. Then, the O_2_ fraction in the anesthetic gas was set to hyperoxic gas challenge conditions of 41%, 71%, and 100% O_2_ using an air–oxygen blender (Sensor Medics Corporation, Yorba Lina, CA, USA). The oxygen levels were chosen to homogenously span the range between normoxic 21% and a maximum of 100% oxygen. For each BOLD measurement, a $T_{2}{}^{*}$ multigradient echo sequence (MGE) was used (TR: 850 ms, TE: 14 echoes from 7–34.47 ms with an interval of 2.11 ms, matrix size: 128 × 128, number of slices: 10; slice thickness: 1 mm, spatial resolution: 0.278 × 0.222 mm, acquisition time for 3 cycles: 4 min). In between each hyperoxic measurement cycle consisting of three $T_{2}{}^{*}$ map acquisitions, a five-minute waiting period was included into the scanning protocol to allow enough accumulation time for oxygen in the blood. At the end of imaging, mice were euthanized by cervical dislocation.

### 2.3. Image Postprocessing

$T_{2}{}^{*}$ and *T*_1_ maps were generated using the image sequence analysis tool in the Paravision 7 software suite. Then, using in-house developed MATLAB code (version 2018a), $T_{2}{}^{*}$ relaxivity ($R_{2}{}^{*}$) was calculated as the inverse of $T_{2}{}^{*}$. Subsequently, $R_{2}{}^{*}$ parameter maps averaged over the three baseline image acquisition cycles were generated. The difference between the averaged baseline $R_{2}{}^{*}$ parameter map and challenged image acquisitions at each cycle was calculated using Equation (1):(1)$$\Delta R_{2}{}^{*}=R_{2}{{}^{*}}_{\left(challenge\right)}-R_{2}{{{}^{*}}_{avg}}_{\left(baseline\right)}$$
where $R_{2}{{}^{*}}_{\left(challenge\right)}$ corresponds to challenged $R_{2}{}^{*}$ parameter maps per oxygen level, while $R_{2}{{{}^{*}}_{avg}}_{\left(baseline\right)}$ relates to the average baseline $R_{2}{}^{*}$ parameter map.

Following the above calculations, to discriminate between hyper-oxidation responsive and non-responsive voxels, all $\Delta R_{2}{}^{*}$ parameter maps were filtered according to Equation (2), as presented in [25]:(2)$$\left|\Delta R_{2}{}^{*}\right|\left\{\begin{array}{c} <2*SD\left(\Delta R_{2}{{}^{*}}_{21\%}\right):\;non-responsive\\ >2*SD\left(\Delta R_{2}{{}^{*}}_{21\%}\right):\;responsive \end{array}\right.$$
where $SD\left(\Delta R_{2}{{}^{*}}_{21\%}\right)$ refers to the standard deviation in $\Delta R_{2}{}^{*}$ in each voxel during the baseline scans. Only responsive voxels were included in subsequent analyses of $\Delta R_{2}{}^{*}$, which were performed in the MITK Workbench (version 2021.10, DKFZ, Heidelberg, Germany). Regions of interest (ROIs) were drawn on the slice including the largest tumor diameter and the least artifacts from breathing motion, while avoiding obvious necrotic tumor regions that were identified on the *T*_1_-weighted anatomical reference scans. The “view” module included in pmod (version 4.303) was used for the visualization of parameter maps.

### 2.4. Statistical Analysis

All statistical analyses were performed in R studio (version 1.2.5033). The differences in $\Delta R_{2}{}^{*}$ between hyperoxic gas challenges within each breast cancer molecular subtype, as well as the differences in $\Delta R_{2}{}^{*}$ between breast cancer molecular subtypes within each hyperoxic gas challenge were calculated using Kruskal–Wallis significance tests combined with Benjamini–Hochberg post hoc corrections to account for multiple testing. For significance testing, *p* < 0.05 was considered statistically significant.

## 3. Results

Due to motion artifacts in *T*_2_*** mapping, *n* = 1 luminal A, *n* = 3 Her2^+^ and *n* = 1 triple-negative breast cancers were excluded from subsequent analyses. Table 1 summarizes the $\Delta R_{2}{}^{*}$ (median, IQR) measurements obtained for luminal A (less aggressive), Her2^+^ (moderately aggressive), and triple-negative (highly aggressive) breast cancer via BOLD-MRI using different hyperoxic gas challenges (41%, 71% and 100% O_2_). When looking at the differences in $\Delta R_{2}{}^{*}$ between hyperoxic gas challenges with each breast cancer molecular subtype, the results show that in all subtypes, $\Delta R_{2}{}^{*}$ decreased with increasing O_2_. However, the difference in $\Delta R_{2}{}^{*}$ between hyperoxic gas challenges was only significant between 41% O_2_ and 100% O_2_ in luminal A breast cancer (*p* = 0.0073).

Meanwhile, independently of the oxygen concentration, significant differences between luminal A and Her2^+^, as well as luminal A and triple-negative breast cancer were observed. At 100% O_2_, these differences became most pronounced. No significant difference between Her2^+^ and triple-negative breast cancer was observed at any oxygen concentration. The characterization of breast cancer molecular subtypes based on $\Delta R_{2}{}^{*}$ using different hyperoxic gas challenges is illustrated in Figure 1.

Figure 2 shows exemplary $\Delta R_{2}{}^{*}$ maps masked onto a corresponding *T*_1_-weighted image for a triple-negative breast cancer xenograft under different hyperoxic gas challenges. The figure shows that from 41%, 71%, to 100% O_2_, the steady decrease in $\Delta R_{2}{}^{*}$ is clearly visible, especially in the dorsal half of the lesion.

## 4. Discussion

The results of our preclinical study show that there is a significant difference between luminal A and Her2^+^, as well as luminal A and triple-negative BC in their response to hyperoxic BOLD-MRI independently of the amount of oxygen. While the significant difference is most pronounced at 100% O_2_., it is already achievable at 41% O_2_.

Breast cancer is known for its high level of intratumoral heterogeneity, where varying selective pressures of the hypoxic tumor microenvironment induce divergent physiological phenotypes between tumor subregions [12,13]. Currently, treatment plans are based on invasive tissue sampling, which is inherently prone to site selection bias and can rarely capture the full extent of intratumoral heterogeneity, contributing to treatment failure and patients’ deaths [14]. It is therefore crucial to establish non-invasive imaging biomarkers that identify molecular subtypes of breast cancer and enable a comprehensive depiction of hypoxic tumor physiology. Recently, for example, Parkins et al. [26] performed non-invasive PET/MRI in a murine model of triple-negative breast cancer. The authors identified PET-based biomarkers that directly link hypoxia to the heterogeneous alternation of PD-L1 expression, highlighting the potential of imaging biomarkers to characterize hypoxia-induced physiological heterogeneity and support treatment planning in breast cancer. MRI also provides other non-invasive contrasting techniques for the assessment of intratumoral hypoxia via the BOLD-effect [15,27,28]. The BOLD effect is dependent on the fraction of oxygenized hemoglobin in the tumor microvasculature, making it a useful diagnostic tool for the identification of oxygen-deprived tumor subregions. BOLD-MRI using hyperoxic gas challenges is performed to introduce diagnostic variation of intratumoral deoxyhemoglobin concentration, thereby allowing the identification of structurally intact tumor regions. Until now, measurements of the BOLD effect have mostly been performed using hyperoxic gas challenges of 100% O_2_ or carbogen (95% O_2_ and 5% CO_2_) [15,22,28]. However, several studies reported that hyperoxic gas challenges, especially with carbogen, were not well tolerated by patients [22,28]. Moreover, the oxygen equilibrium curve suggests that the amount of oxygenized hemoglobin barely changes beyond pO_2_ = 70 mmHg, which corresponds to normoxic conditions of 21% O_2_ in the breathing air [24]. We therefore hypothesized that molecular subtype characterization of breast cancer with BOLD-MRI using hyperoxic gas challenges of less than 100% O_2_ would lead to similar results as when 100% O_2_ is used. Of note, the ability to use less than 100% O_2_ would greatly improve the technique’s applicability in standard clinical practice.

Indeed, our results show that an increase to 100% O_2_ had no beneficial effect on the characterization of breast cancer. Instead, 71% O_2_ or even 41% O_2_ were sufficient to detect measurable decreases in the transverse relaxation rate ($\Delta R_{2}{}^{*}$). Across all $\Delta R_{2}{}^{*}$ measurements, statistically significant differences between less aggressive luminal A and moderately aggressive Her2^+^, as well as between luminal A and highly aggressive triple-negative breast cancer were observed. On the other hand, no significant difference in $\Delta R_{2}{}^{*}$ measurements between Her2^+^ and triple-negative breast cancer was observed. Further, under all oxygen levels used in this study, $\Delta R_{2}{}^{*}$ was the lowest in luminal A breast cancer. As for the lack of distinction between Her2^+^ and triple-negative breast cancer, such lack has been observed previously in studies using dynamic-contrast-enhanced MRI [29,30,31]. As for the low $\Delta R_{2}{}^{*}$ measurements in luminal A breast cancer, this can be explained by the fact that the magnitude of BOLD signal decrease following a hyperoxic gas challenge depends on the vascular maturity in the tissue, i.e., the ability of vessels to transport oxygen-enriched blood [32,33], and clinical studies using contrast-enhanced MRI have shown that luminal A tumors develop the most functional blood vessels in comparison to Her2^+^ and triple-negative tumors [34].

To our knowledge, this is the first preclinical study to successfully discriminate luminal A from other breast cancer subtypes using hyperoxic BOLD-MRI. Previously, Virani et al. conducted a preclinical study including a hyperoxic gas challenge and found correlations between BOLD parameters and tumor hypoxia in mouse glioblastoma models [35]. In another preclinical study, McPhail and Robinson [36] found correlations between carbogen-induced $\Delta R_{2}{}^{*}$ and the extent of hypoxia in chemically induced rat breast tumors. Similarly, Yang et al. [25] found associations between histologically-proven hypoxic tumor subregions and BOLD-MRI signal following a 100% O_2_ challenge in rat breast tumors. Others have used BOLD-MRI measurements to predict treatment response in rat models of breast cancer [37] and prostate cancer [38,39].

Studies in humans have found correlations between $R_{2}{}^{*}$ measured at 21% O_2_ and tumor hypoxia in patients with invasive ductal carcinoma [40,41]. Using a hyperoxic gas challenge, Rakow-Penner et al. [28] found that in a preliminary study BOLD-MRI allows for the discrimination of benign and malignant breast lesions from healthy breast tissue. However, so far, attempts to differentiate benign from malignant breast lesions have not been successfully performed [42,43]. The study by Liu et al. [40] showed that there was no correlation of tumor $R_{2}{}^{*}$ at 21% O_2_ and breast cancer receptor status, highlighting the complexity of breast cancer characterization based on BOLD parameters.

Indeed, the lack of association between BOLD signal and receptor status in previous studies may be explained by the low field strengths used in clinical routine. Because of the field dependency of $R_{2}{}^{*}$ [44,45], changes in $R_{2}{}^{*}$ following a hyperoxic gas challenge would be less pronounced at 1.5 Tesla or 3 Tesla compared to the 9.4 Tesla system used in our study. The increased signal-to-noise ratio at high field strengths allow higher resolution scans to be obtained, which leads to less disrupting partial volume effects. This provides more detailed information on tumor physiology, especially in highly heterogenous cancers such as breast cancer [11,13].

Our results add to the characterization of molecular subtypes of breast cancer with hyperoxic BOLD-MRI and without the use of intravenous Gadolinium-based contrast agents, which have been under debate over their potential side-effects and toxicity [46]. Furthermore, the results of our study suggest that the adverse effects during carbogen breathing during BOLD-MRI in the clinical setting can be reduced by setting the oxygen level to 41%. This may significantly improve the translatability of hyperoxic gas challenges while reducing the adverse effects of oxygen toxicity during prolonged oxygen breathing and promote the introduction of $\Delta R_{2}{}^{*}$ as a promising contrast agent-free imaging biomarker of breast cancer.

Our study has some practical limitations, which include its preclinical nature using a limited number of luminal A, Her2^+^, and triple-negative tumor xenografts only. We did not quantify the fraction of responsive voxels for each BC subtype, which may prove to be useful to the characterization of BCs based on hyperoxic BOLD-MRI. Future translational studies should test whether molecular subtype characterization of breast cancer with BOLD-MRI using hyperoxic gas challenges of less than 100% O_2_ is feasible and useful in humans. In addition, hyperoxic gas challenges using lower levels of oxygen may also be of value to differentiate benign lesions (such as fibroadenomas) from malignant breast lesions, which was not within the scope of the study.

## 5. Conclusions

Hyperoxic BOLD-MRI enables a distinction between luminal A and Her2^+^ and between luminal A and triple-negative breast cancer independently of oxygen level in a hyperoxic gas challenge. This molecular subtype characterization was already achieved with 41% O_2_. Overall, we conclude that hyperoxic BOLD-MRI without the use of intravenous contrast agents is promising to provide imaging biomarkers for breast cancer characterization and the assessment of intratumoral heterogeneity. Future studies could focus on the verification of our results in a clinical setting via the implementation of hyperoxic BOLD-MRI into multiparametric MRI protocols.

## Figures and Tables

**Figure 1 diagnostics-13-02946-f001:**
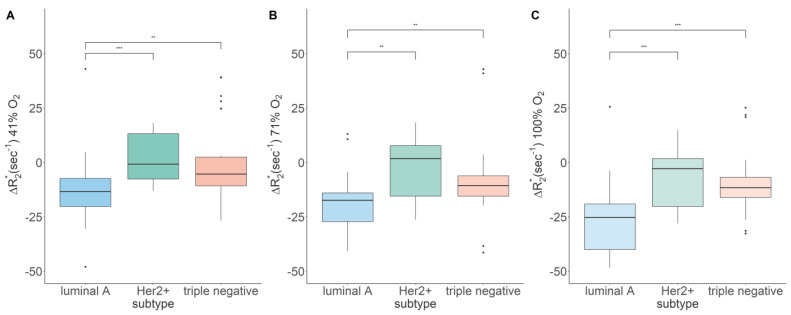
Boxplots showing the differences in response to the hyperoxic gas challenges between breast cancer molecular subtypes (MCF-7: *n* = 9; SKBR-3: *n* = 5; MDA-MB-231: *n* = 9; acquisition cycles per oxygenation level: *n* = 3; ** *p* < 0.01, *** *p* < 0.001). Boxplots correspond to (**A**) 41%, (**B**) 71%, and (**C**) 100% O_2_.

**Figure 2 diagnostics-13-02946-f002:**
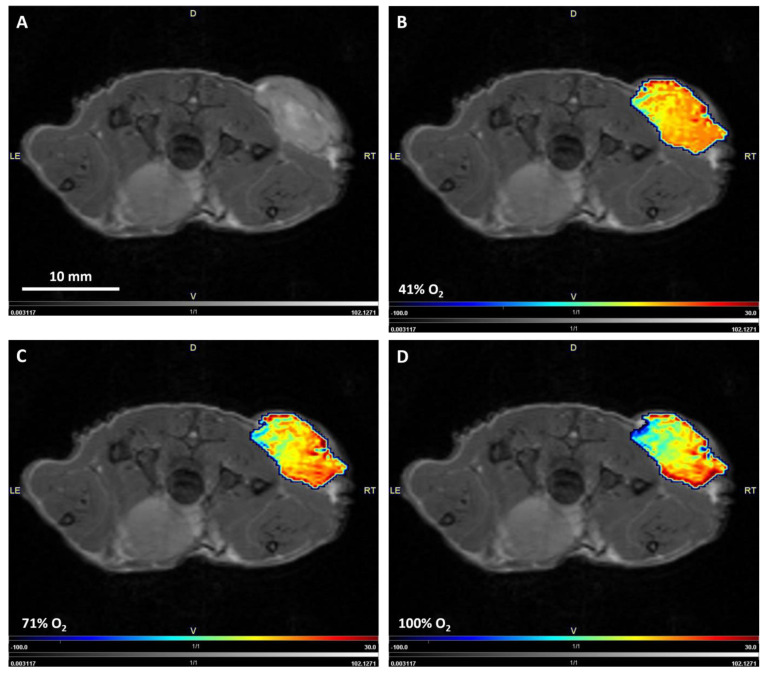
*T*_1_ weighted image of a triple-negative breast cancer in axial view (**A**) and exemplary $\Delta R_{2}{}^{*}$ maps at 41% (**B**), 71% (**C**), and 100% (**D**) O_2_, which were overlaid onto the *T*_1_-weighted anatomical reference image. D = dorsal, V = ventral, LE = left, RT = right.

**Table 1 diagnostics-13-02946-t001:** Overview of $R_{2}{}^{*}$ (median, IQR) at baseline (i.e., 21% oxygen), as well as $\Delta R_{2}{}^{*}$ measurements from BOLD-MRI using different hyperoxic gas challenges (median, IQR) obtained for luminal A, Her2^+^, and triple-negative breast cancer.

	21% O_2_ [s^−1^]	41% O_2_ [s^−1^]	71% O_2_ [s^−1^]	100% O_2_ [s^−1^]
**Luminal A**	145.19, 67.56	−13.46, 14.20 ^a^	−20.37, 17.06	−28.42, 24.35 ^a^
**Her2^+^**	140.60, 33.13	−0.76, 20.80 ***	1.80, 23.19 **	−2.85, 21.95 ***
**Triple negative**	110.06, 20.93	−5.32, 13.15 **	−10.58, 9.38 **	−11.64, 9.15 ***

^a^ within a row: values with the same superscript differ significantly (*p* < 0.05), * within a column: asterisks indicate significant differences from luminal A; (** *p* < 0.01, *** *p* < 0.001).

## Data Availability

The data presented in this study are available on request from the corresponding author.

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
