# Peer review of "Hyperoxic BOLD-MRI-Based Characterization of Breast Cancer Molecular Subtypes Is Independent of the Supplied Amount of Oxygen: A Preclinical Study"

_diagnostics, 2023, doi:10.3390/diagnostics13182946_

Round 1

Reviewer 1 Report

Hypoxia is known to be a poor prognostic indicator for nearly all solid tumors. Imaging has potential to identify, spatially map and quantify tumor hypoxia prior to therapy, as well as track changes in hypoxia on treatment. At present no hypoxia imaging methods are available for routine clinical use but there is gathering evidence that MRI techniques may provide a practical and more readily translational alternative. Therefore, the authors of the paper, made an attempt of the molecular subtype of breast cancer distinction in mice, using hyperoxic BOLD-MRI and showed, that it could be achieved  independently of the amount of oxygen.

The reviewed work is clear, relevant for the field and presented in a well-structured manner (include introduction, materials and methods section, separately described results, discussion and conclusions). The cited references are mostly recent, relevant publications and an excessive number of self-citations was not included in the manuscript. The manuscript is scientifically sound and the experimental design is appropriate to test the hypothesis. The methods used in the present study are well described and justified by proper references. The data is interpreted appropriately and consistently throughout the manuscript The conclusions are consistent with the evidence and presented arguments. Adequate ethics permissions were obtained.

There are only some minor concerns related to the appearance of the figures, that need to be addressed:

1.     Figure 1 - The figure should be presented in other way , e.g. smaller panels could be used and all of them could be presented in one row or maybe authors have additional results which could by used as a forth panel ? Alternatively, the more extended figure could be interesting, including 6 panels in 3 rows and 2 columns, where first column include boxplots and the second column include exemplary ∆R2* maps at chosen oxygen level - 41% , for three subtypes of breast cancer.

2.     Figure 2 - The figure should be a little more self explained. Oxygen levels should be included in images e.g near the subfigure labels.

Author Response

Comment 1: Figure 1 - The figure should be presented in other way , e.g. smaller panels could be used and all of them could be presented in one row or maybe authors have additional results which could by used as a forth panel ? Alternatively, the more extended figure could be interesting, including 6 panels in 3 rows and 2 columns, where first column include boxplots and the second column include exemplary ∆R2* maps at chosen oxygen level - 41% , for three subtypes of breast cancer.

Answer 1: We have revised figure 1 and rearranged the panels for better comparability between BC subtypes and oxygen levels.

Comment 2: Figure 2 - The figure should be a little more self explained. Oxygen levels should be included in images e.g near the subfigure labels

Answer 2: We have included the oxygen levels in the subfigure labels.

Reviewer 2 Report

The research is vey useful thanks for all contributions.

Author Response

We thank the reviewer for his comment on our manuscript.

Reviewer 3 Report

Overall, the topic of this manuscript is interesting, and most of manuscripts are clear and the results are sound. But there are several minor places that need to be improved.

1. In line 119 page 3, why are the numbers of the mice in three breast cancer groups different? Her2+ is 4 mice less than other two groups. Will this affect the results?

2. For the cancer growth, the authors only mentioned "suitable size", how large the cancer growth? the author should give the ranges

3. In line 179, page 4, the author mention "luminal A (less aggressive), Her2+ (moderately aggressive), and triple negative (highly 179 aggressive)", the three groups represent different levels of aggressive breast cancer. But those information are very important to understand this research. the author should mention this information earlier and in abstract.

4.  Why did the author only select 41%, 71% and 100%? Whether are there any other influences if using smaller scales?

5. In Fig. 2, what are the color values standing for to be more blue and to be more red. And also, how about the scale bar?

Author Response

Comment 1: In line 119 page 3, why are the numbers of the mice in three breast cancer groups different? Her2+ is 4 mice less than other two groups. Will this affect the results?

Answer 1: We have specified the sample sizes before the exclusion of datasets due to motion artifacts in T2* mapping:

Page 3, line 119: “(…) luminal A (n = 10), Her2+ (n = 8), or triple negative (n = 10) (…)”

Page 4, line 181f: “Due to motion artifacts in T2* mapping, n = 1 luminal A, n = 3 Her2+ and n = 1 triple negative breast cancers were excluded from subsequent analyses.”

Comment 2: For the cancer growth, the authors only mentioned "suitable size", how large the cancer growth? the author should give the ranges

Answer 2: We have specified that tumors were allowed to grow to a maximum diameter of 10mm:

Page 3, line 124f: “(…) with a maximal diameter of 10 mm.”

Comment 3: In line 179, page 4, the author mention "luminal A (less aggressive), Her2+ (moderately aggressive), and triple negative (highly 179 aggressive)", the three groups represent different levels of aggressive breast cancer. But those information are very important to understand this research. the author should mention this information earlier and in abstract.

Answer 3: The information is given in the Material and Methods section of the manuscript on page 3, lines 109-111.

Comment 4: Why did the author only select 41%, 71% and 100%? Whether are there any other influences if using smaller scales?

Answer 4: We have specified that we chose the oxygen levels to homogenously span the range of normoxic (21%) to 100% oxygen in the anesthetic gas:

Page 3, line 145f: “The oxygen levels were chosen to homogenously span the range between normoxic 21% and a maximum of 100% oxygen.”

Comment 5: In Fig. 2, what are the color values standing for to be more blue and to be more red. And also, how about the scale bar?

Answer 5: We have adapted figure 2 to include a scale bar and changed the figure caption to clarify that the color values correspond to  maps masked onto the anatomical reference images.